# Hyaluronic acid hydrogels support to generate integrated bone formation through endochondral ossification in vivo using mesenchymal stem cells

**Shintaro Yamazaki[1,2], Ryoko Hirayama[1], Yayoi Ikeda[3], Sachiko Iseki[1], Tetsuya Yoda[2], Masa-Aki Ikeda[1]** *

1 Department of Molecular Craniofacial Embryology, Graduate School of Medical and Dental Sciences, Tokyo Medical and Dental University, Tokyo, Japan, 2 Department of Maxillofacial Surgery, Graduate School of Medical and Dental Sciences, Tokyo Medical and Dental University, Tokyo, Japan, 3 Department of Anatomy, Aichi Gakuin University School of Dentistry, Nagoya, Aichi, Japan

* mikeda.emb@tmd.ac.jp

**Data Availability Statement:** All relevant data are within the paper and its Supporting information files.

## Abstract

Engineered cartilage tissue from differentiated mesenchymal stem cells (MSCs) can generate bone in vivo through endochondral ossification (ECO). This ECO-mediated approach has the potential to circumvent the severe problems associated with conventional MSC-based bone tissue engineering techniques that lack mechanisms to induce angiogenesis. Hyaluronic acid (HA) is a key component in the cartilage extracellular matrix. However, the ECO-supporting properties of HA remain largely unclear. This study aimed to compare the ability of HA and collagen hydrogels to support in vitro differentiation of MSC-based hypertrophic cartilage tissues and to promote endochondral bone formation in vivo. Following the chondrogenic and hypertrophic differentiation in vitro, both HA and collagen constructs accumulated sulfated glycosaminoglycan (sGAG) and type 1, type II, and type X collagen. However, HA hydrogels exhibited a more uniform distribution of sGAG, type 1 collagen, type X collagen, and osteocalcin proteins; in addition, the cells embedded in the hydrogels had more rounded cell morphologies than those in the collagen constructs. At week 5 of in vitro culture, two to three constructs were implanted into a subcutaneous pocket in nude mice and harvested after 4 and 8 weeks. Both HA and collagen constructs promoted endochondral bone formation with vascularization and bone marrow development; however, the HA constructs fused to form integrated bone tissues and the bone marrow developed along the space between the two adhered grafts in all implanted pockets (n = 5). In the collagen constructs, the integration was observed in 40% of the pockets (n = 5). Microcomputer CT analysis revealed that the bone volume of HA constructs was larger than that of collagen constructs. In conclusion, compared to collagen hydrogels, HA hydrogels had superior potential to generate integrated bone with vascularization and bone marrow development. This study provides valuable insights for applying ECO-mediated bone tissue engineering approaches for the repair of critical-sized bone defects.

**Funding:** This work was supported by a Grant-in-Aid for Scientific Research (KAKENHI) from the Japan Society for the Promotion of Science (JSPS) (grant nos. JP19K10259 and 22K10032 to MAI), i.e., KAKENHI means "Grant-in-Aid for Scientific Research" in Japanese. The funders had no role in study design, data collection and analysis, decision to publish, or preparation of the manuscript.

**Competing interests:** The authors have declared that no competing interests exist.

## Introduction

Autogenous bone is the gold standard for repairing bone defects resulting from oral trauma, congenital defects, or surgical resection. However, autografting has serious limitations, including morbidity and limited bone volume at the donor site [1–4]. Tissue engineering has gained attention as an alternative strategy for the repair/regeneration of bone defects. In maxillofacial surgery and orthopedic bone augmentation, traditional cell-based bone tissue engineering applications aim for direct bone formation by mimicking intramembrane ossification (IMO); mesenchymal stem cells (MSCs) seeded on a support scaffold directly differentiate into osteogenic cells [5]. However, the lack of an angiogenic step in the IMO process results in inadequate angiogenesis of the graft after transplantation in vivo [6–8], resulting in the inapplicability of this approach to repair large bone defects.

Strategies aimed at recapitulating innate mechanisms in skeletal development and fracture healing can overcome this challenge. In bone formation and fracture healing via endochondral ossification (ECO), chondrocytes in a developing cartilage template undergo hypertrophy and release proangiogenic factors. This promotes the infiltration of blood vessels, while the cartilage template remodels into bone [9]. The ECO-mediated approach using tissue-engineered cartilage constructs avoids the critical problems associated with traditional IMO-based methods. This is attributed to the ability of chondrocytes to tolerate hypoxic conditions, release proangiogenic factors, and transform an avascular cartilage graft into angiogenic tissue in the ECO processes. Implementing such an ECO program will enable MSC-based cartilage grafts to generate bone in vivo [10–18].

An important requirement for the clinical application of this ECO-mediated approach is fabricating the desired amount of cartilage grafts. It is not practical to clinically prepare cartilage of a size that fits the bone defect; therefore, the cartilage to be transplanted should form bone integrally, when multiple pieces are implanted [19]. Hydrogels may be an attractive tool for scaling up tissue-engineered grafts for endochondral bone ossification. Many naturally derived hydrogels support MSC chondrogenesis in vitro and ECO in vivo [20–29]; however, the optimal support material that meets the requirements for clinical application should be determined.

Hyaluronic acid (HA) is a biodegradable and highly biocompatible polysaccharide present in the extracellular matrix of cartilage [30]. HA interacts with MSCs via surface receptors such as CD44 and supports chondrogenic differentiation [22, 23, 25, 27–29, 31]. Furthermore, HA scaffold promotes IMO-mediated bone differentiation of human dental pulp stem cells [32], and the scaffold combined with collagen promotes ECO-mediated bone formation [33, 34]. However, the ECO-supporting properties of hydrogels containing only HA remain unclear.

This study aimed to evaluate HA hydrogels for supporting hypertrophic cartilage formation in vitro and the subsequent endochondral ossification in vivo, using bone marrow-derived adult human MSCs. We compared the properties of HA with those of collagen, which is widely applied for MSC-based bone tissue engineering and is an effective material for scaling up engineered grafts for endochondral bone ossification [14]. Human MSC-seeded HA and collagen constructs were evaluated for in vivo ECO potential through subcutaneous implantation in an immunodeficient mouse model. Our results revealed that hyaluronic acid hydrogel has excellent properties as a scaffold for MSCs to scale up engineered grafts to generate integrated bone through endochondral bone ossification.

## Materials & methods

### Expansion of human MSCs

Primary human bone marrow-derived MSCs (Lonza Japan, Tokyo) at passage two were expanded with MSCG Mesenchymal Stem Cell Growth Medium (Lonza Japan) at 37˚C in 5%

$CO_2$ and 95% humidified air, according to the manufacturer's instructions. When cells reached confluence, they were harvested and cryopreserved at -80°C, in liquid nitrogen. The resulting frozen stocks (passage 3) were used as starting cell sources for this study. The chondrogenic potential of MSCs was assessed through micromass culture; two donors with high potential were selected for the following experiments. For the experiments, the frozen MSCs were thawed, seeded at a density of $5 \times 10^3$ cells/cm, and then expanded to passage 4 (referred to as undifferentiated MSC) in Dulbecco's Modified Eagle Medium and Ham's F12 (DMEM/F-12) medium (Sigma-Aldrich) containing 10% fetal bovine serum (FBS) (Hyclone) supplemented with 50 μg /mL gentamicin, 200 M/mL L-alanyl-L-glutamine (both FUJIFILM Wako Pure Chemical, Japan), and 1 ng /mL basic fibroblastic growth factor (bFGF, Reprocell).

## Preparation of MSCs-seeded hydrogels

To generate MSC-encapsulating hyaluronic acid (HA) hydrogels, HyStem hydrogel (Sigma-Aldrich) was used. Briefly, Glycosil (thiolated hyaluronan) and Extralink, a thiol-reactive crosslinker, polyethylene glycol diacrylate (PEGDA), were dissolved in degassed water (10 mg/mL). MSC suspension was mixed with HA solution composed of Glycosil and Extralink at 1:4 (v/v), on ice. Droplets of MSCs-containing HA solution (15 μL, $2.5 \times 10^5$ cells per droplet) were quickly deposited onto a paraffin-coated 24-well plate and allowed to gelate at 37°C for 30 min. To generate MSC-encapsulating collagen hydrogels, MSC suspension was mixed with a neutralized collagen solution composed of Cellmatrix Type I-A, 10× Minimum Essential Medium (MEM), and 10× Reconstitution buffer (50 mM NaOH, 260 mM NaHCO3, 200 mM HEPES) (Nitta Gelatin, Osaka, Japan) at a ratio of 8:1:1 (V/V/V), on ice. Droplets of MSC-containing collagen solution (30 μL, $5 \times 10^5$ cells per droplet) were deposited onto a paraffin-coated 24-well plate and gelated at 37°C for 30 min.

## In vitro differentiation conditions

Droplets of MSC-seeded HA and collagen hydrogels were cultured in a chondrogenic differentiation medium for 3 weeks. The chondrogenic medium consisted of high-glucose Dulbecco's modified Eagle's medium (DMEM) (Sigma-Aldrich) supplemented with 1% ITS-G supplement, 0.12% bovine serum albumin, 50 μg/mL gentamicin, 0.35 mM L-proline, (all from FUJIFILM Wako Pure Chemical), 100 nM dexamethasone (Sigma-Aldrich), and 10 ng/ml TGF-ß3 (Prospec-Tany TechnoGene, Israel). After 3 weeks, droplets were either subjected to analysis or cultured in a hypertrophic differentiation medium for 2 weeks. The hypertrophic medium consisted of high glucose DMEM supplemented with 10 mM ß-glycerophosphate, 0.12% bovine serum albumin, 10 nM dexamethasone, 200 μM ascorbate-2-phosphate, 50 nM thyroxine (all from Sigma-Aldrich), 50 μg/mL gentamycin, and 50 pg/mL IL-1ß (PeproTech).

## Animals

Four-week-old male nude mice (BALB/cSlc-nu/nu, Sankyo Labo Service Corporation, INC., Japan) were used in this study. Three or four mice were housed in a cage under a standard 12-h light/dark cycle (light on at 8:00 and off at 20:00) at 22–24°C and 20–70% relative humidity, with free access to food and water. Autoclaved bedding was provided for each cage and changed once a week. All animal experiments were conducted in accordance with the guidelines approved by the Institutional Animal Care and Use Committee of Tokyo Medical and Dental University (approval ID: A2019-204C, A2020-116A, and A2021-121A). This study followed the Animal Research: Reporting of In Vivo Experiments (ARRIVE 2.0) guidelines.

## In vivo implantation of MSCs

Mice were anesthetized using a mixture of three drugs: 0.75 mg/kg body weight (b.w.) medetomidine (Domitor, Nippon Zenyaku Kogyo Co., Ltd., Tokyo, Japan), 4.0 mg/kg b.w. midazolam (Dormicum, Astellas Pharma Inc., Tokyo, Japan), and 5.0 mg/kg b.w. butorphanol (Vetorphale, Meiji Seika Kaisha, Ltd., Tokyo, Japan). These drugs were diluted in sterile saline to 0.1 ml/10 g b.w./mouse and were administered through intraperitoneal injection. HA and collagen constructs that differentiated in the hypertrophic medium for 2 weeks were implanted subcutaneously into the back of nude mice. Briefly, two subcutaneous pockets were created along the central line of the spine in each mouse, and two to three constructs were inserted into each pocket (n = 5). In some experiments, to prevent hyaluronan constructs from fusing, four pockets were created in both lateral sites of the spine, two at the shoulders and two at the hips; one hyaluronan construct was inserted into each pocket (4 constructs/mouse). After suturing the surgical sites, mice were injected with an antagonist: 0.75 mg/kg b.w. atipamezole (Antisedan; Nippon Zenyaku Kogyo Co., Ltd.). The animals were monitored every couple of days during the healing period for any possible complications. The number of experimental samples was determined based on earlier reports [10–14, 17, 18]. Mice of the same age were randomly selected and assigned to each implantation group. A total of 23 mice were used in the in vivo transplantation studies, including preliminary studies to determine the optimal amount of hydrogel and the number of MSCs.

The mice were euthanazed by carbon dioxide inhalation with little suffering at 4 and 8 weeks post-implantation, and the constructs were collected for analysis.

## Quantitative analysis of sulfated Glycosaminoglycan (sGAG)

The sGAG content was measured 3 weeks after in vitro chondrogenic differentiation using the Blyscan glycosaminoglycan assay (Biocolor, UK). Briefly, the constructs were digested with papain overnight at 65˚C. The amount of sGAG was quantified using Blyscan dye reagent according to the manufacturer's instructions. The DNA concentration in the supernatant was determined using the Quanti-iT dsDNA HS assay kit (Invitrogen). Sulfated GAG content was normalized to DNA content.

## Alkaline phosphatase (ALP) activity

ALP activity was measured 2 weeks after hypertrophic differentiation (a total of 5 weeks of in vitro culture). Constructs were sonicated on ice in lysis buffer containing 1% NP-40. ALP activity in the supernatant was analyzed using a Lab assay ALP kit (Fujifilm Wako Pure Chemical, Osaka, Japan). ALP activity was normalized to DNA content (as described above).

## RNA extraction and real-time PCR analysis

Constructs were sonicated in Tri Reagent (Sigma-Aldrich). The total RNA was purified using a ReliaPrep RNA Tissue Miniprep System (Promega), according to the manufacturer's instructions. Complementary DNAs were synthesized using the ReverTra Ace qPCR RT Master Mix with gDNA remover (Toyobo, Osaka, Japan). Quantitative real-time RT-PCR analysis was performed in triplicate using the LightCycler 480 instrument (Roche Applied Science, Indianapolis, IN, USA) and GoTaq qPCR Master Mix (Promega), according to the manufacturer's instructions. The primer sequences used for RT-PCR are described in S1 Table. Expression of all genes was normalized to that of the tyrosine 3-monooxygenase/tryptophan 5-monooxygenase activation protein zeta (YWHAZ) gene [35]. The relative expression of mRNA was calculated using the $2^{-\Delta\Delta Cq}$ method [36].

## Western blot analysis

Constructs were frozen in liquid nitrogen, ground into fine particles, resuspended in 30–50 mL Laemmli sample buffer, sonicated on ice, and then boiled for 10 min, after which the DNA concentration of the samples was determined. Proteins standardized by DNA content were separated on 7.5% or Any kD Mini-PROTEAN TGX precast polyacrylamide gels (Bio-Rad) and transferred to PVDF membranes (Bio-Rad). The membranes were probed with the following primary antibodies: rabbit polyclonal type I (1:1000, Proteintech), type II collagens (1:1000, Santa Cruz Biotechnology), α-Tubulin (1:1000, MBL), GAPDH (1:1000, GeneTex), and goat polyclonal aggrecan (1:1000, R&D Systems) antibodies. After multiple washes, the membranes were incubated with HRP-conjugated anti-rabbit or anti-goat IgG (eBioscience). After multiple washes, the membranes were treated with the Amersham ECL Prime Western Blotting Detection Reagent (Cytiva). Images of bound antibodies were digitally captured using a MicroChemi chemiluminescent system (DNR Bio-Imaging Systems, Israel), and band intensities were quantified using GelQuant Analysis Software (DNR Bio-Imaging Systems).

## Histological and immunohistochemical analysis

Following the in vitro and in vivo studies, the constructs were fixed in 4% paraformaldehyde for 24 h. The in vivo constructs were decalcified using EDTA for up to 3 weeks. The constructs were then embedded in paraffin and sectioned at 6 mm thickness. After rehydration, in vitro sections were stained with safranin O/Fast Green, Alizarin Red, and Von Kossa; the in vivo sections were stained with hematoxylin and eosin (H&E) and tartrate-resistant acid phosphatase (TRAP). For immunohistochemical analysis, in vivo sections were treated with 3% hydrogen peroxide to quench endogenous peroxidase activity. For antigen retrieval, the sections were incubated with 5 mg/ml hyaluronidase for 30 min at 37˚C, followed by digestion with 1 mg/mL pepsin in 0.5 M acetic acid for 30 min at 37˚C, after which the sections were incubated in blocking solution (5% Bloc-Ace, Dainippon Sumitomo Pharma, Osaka, Japan) for 30 min at RT. The sections were then incubated overnight with the following primary antibodies: rabbit polyclonal type I (1:400, Proteintech), type II (1:100, Santa Cruz Biotechnology), type X (1:400, Cloud-Clone Corp.) collagens, osteocalcin (1:400, Proteintech), CD31 (1:100, Abcam), SOX-9 (1:100, Santa Cruz Biotechnology), mouse monoclonal type X collagen (1:100, eBioscience), MMP-13 (1:100, Santa Cruz Biotechnology) or normal rabbit IgG (1:100, Santa Cruz Biotechnology). Peroxidase-conjugated Histofine Simple Stain MAX PO (MULTI) or MAX-PO (R, Nichirei, Tokyo, Japan) was used for secondary antibodies. The signal was detected using the ImmPACT NovaRED Substrate kit (Vector). The circularity of cells was quantified by measuring the number of cells in five different areas (200 μm × 200 μm) each, for the periphery and center of constructs.

## Microcomputed tomography (μCT)

Constructs were scanned using a micro-focus X-ray computed tomography system (applied voltage 90 kV; current 30 μA) (Inspexio SMX-100CT, Shimadzu, Japan). Image data were reconstructed using image analysis software (TRI/3D-BON; Ratoc System Engineering, Tokyo, Japan) to evaluate the bone mineral density (BMD) and volume to obtain 3-D images.

## Statistical analysis

Results are presented as means ± standard deviation (SD). Statistical analysis was performed using GraphPad Prism7 software (GraphPad Software, USA). The Student's *t*-test or one-way ANOVA followed by Tukey's multiple comparisons test were accustomed to determine

significant differences between groups. $^*<0.05$ and $^{**}<0.01$ indicate significant difference between two groups.

## Results

### Chondrogenic and hypertrophic differentiation in vitro

**Biochemical and gene expression analysis.** MSC-seeded constructs in HA and collagen hydrogels were cultured in a serum-free medium containing TGF-β3, an inducer of chondrogenic differentiation. Soon after the culture started, the collagen hydrogels began to shrink, while the HA hydrogels remained unchanged. To reduce the size difference between the two constructs formed after in vitro culture, the number of MSCs and the gel volume in HA hydrogels was halved compared to that of collagen hydrogels. However, the wet weight of the HA constructs was four times heavier than that of the collagen constructs after 3 weeks of chondrogenic culture (P < 0.05) (Fig 1A). The amount of sGAG per construct was similar between the two constructs (Fig 1B); however, the amount of sGAG per DNA was significantly higher in the collagen constructs than that in the HA constructs (2.8-fold, P < 0.05) (Fig 1C). At week 3 of chondrogenic differentiation, the constructs were further induced to mature into hypertrophic cartilage in a medium lacking TGF-β3 and supplemented with β-glycerophosphate

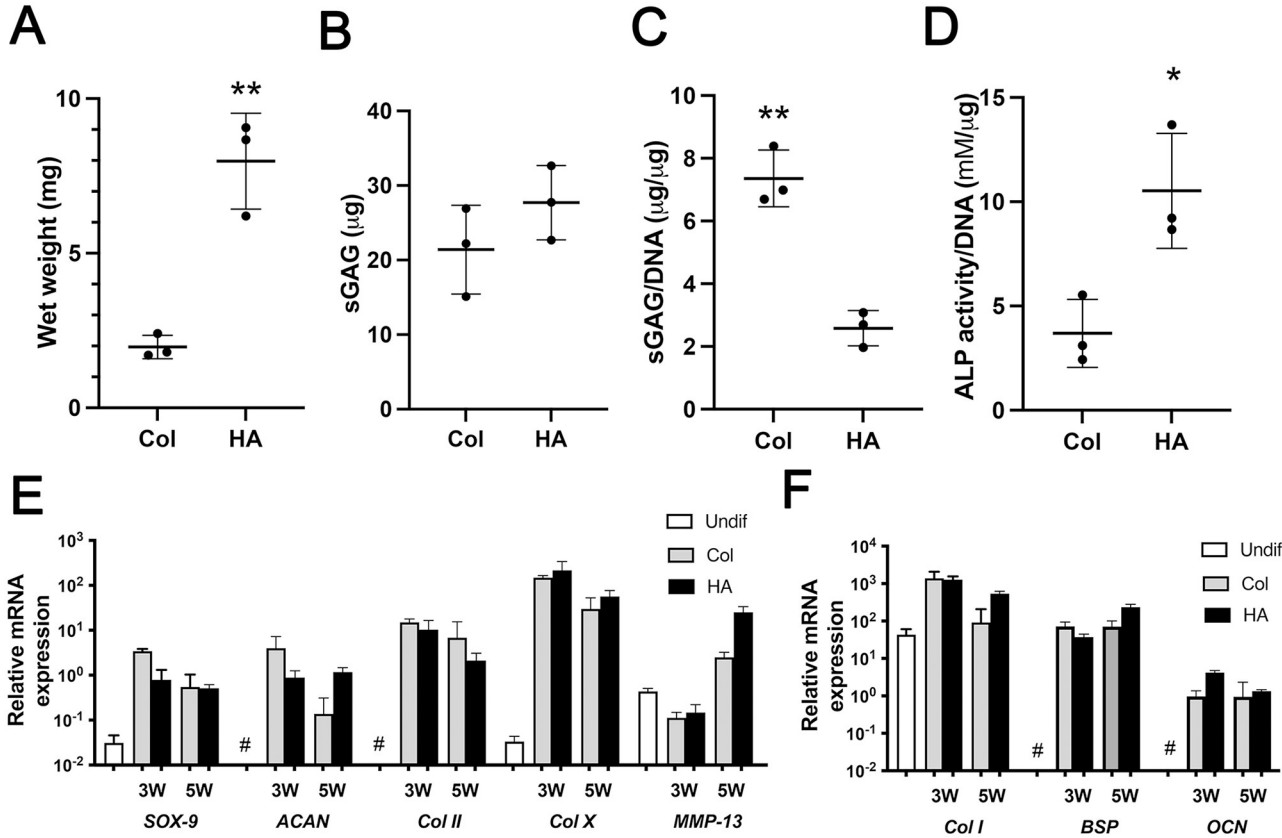

**Fig 1. Biochemical and gene expression analysis of HA and collagen constructs after in vitro culture.** (A) Mass and (B) sGAG accumulation of the constructs, and (C) sGAG accumulation of the constructs normalized to DNA content at week 3 of in vitro culture (n = 3). (D) ALP activity of the constructs normalized to DNA content at week 5 of in vitro culture (n = 3). * indicates significant differences; p < 0.05. (E and F) Gene expression analysis of HA and collagen constructs at 3 and 5 weeks of in vitro culture. Values are presented as mean ± SD. Undifferentiated MSCs expressed high levels of Col I and MMP-13 and low levels of Sox-9 and Col X; they did no express (#) ACAN, Col II, BSP, and OC.

(β-GP) for an additional 2 weeks (total 5 weeks in vitro culture). The ALP activity of HA constructs was significantly higher than that of collagen constructs (2.9-fold, $P < 0.05$) (Fig 1D). The progression of chondrogenic and hypertrophic differentiation during in vitro culture was confirmed through quantitative real-time RT-PCR; chondrogenic (Sox-9, ACAN, and Col II), hypertrophic (Col X and MMP-13), and osteogenic (Col I, BSP, and OC) markers were expressed in both HA and collagen constructs (Fig 1E). Western blot analysis confirmed the relative expression of Col I, Col II, and ACAN proteins (S1 Fig), and immunohistochemical analysis detected the expression of Col X, OCN (Fig 2), MMP-13, and Sox-9 (S2 Fig) proteins in both constructs.

**Histological and immunohistochemical analysis.** At week 3 of in vitro culture, sGAG production (safranin O/Fast Green staining; Fig 2) and the presence of large cells in the lacunae embedded in type II collagen-positive matrix were detected in both constructs, indicating that both HA and collagen hydrogels supported chondrogenesis. However, there were differences in the distribution of sGAG and type II collagen between the two constructs: compared to the collagen constructs, HA constructs showed a homogeneous distribution of sGAG and type II collagen in the engineered tissue. Differences between the two constructs were also evident in cell morphology. In particular, cell morphologies in the periphery varied from irregular/stretched to rounded morphologies in collagen constructs (Fig 2A). In contrast, HA constructs contained cells with more rounded morphologies. The average circularity values of HA constructs were higher than that of collagen constructs in both peripheral and central regions (34% vs. 77.6% (p<0.01) and 78.3% vs. 100% (p<0.05), respectively) (Fig 2B).

Differences between the two constructs were also evident in the distribution of expression of hypertrophic markers at 5 weeks of culture. The increased distribution of type I and X collagen and osteocalcin (OCN) was limited to the outer edge in the collagen constructs. A more homogeneous distribution, except in the case of type I collagen, was observed in the HA constructs. However, calcium deposition was limited to the outer edges in both constructs (Alizarin Red and Von Kossa; Fig 2C and 2D).

## Endochondral bone formation in vivo

**Histological analysis.** At week 5 of in vitro culture, two to three constructs were implanted into a subcutaneous pocket created in the back of nude mice and harvested after 4 and 8 weeks. The HA constructs adhered to each other in all five pockets (Table 1). In the collagen constructs, adhesion was observed in three out of five pockets; the grafts were present independent of each other in the remaining two pockets.

Safranin O and H&E staining assessed spatial cartilage and bone tissue formation post-implantation. At 4 weeks post-implantation, both HA and collagen constructs showed a marked decrease in sGAG staining in the inner cartilage regions and in the formation of osteoid tissues with lamellar morphology in the outer regions (Fig 3D and 3L). At 8 weeks, the HA constructs lost the cartilage phenotype with complete loss of sGAG staining and exhibited strong pericellular staining with eosin in the inner regions (Fig 3O and 3P). The bone marrow component, consisting of a mixture of hematopoietic cells and adipose tissue, developed between the inner cartilage and outer osteoid tissues in both HA and collagen constructs (Fig 3F and 3N, arrowheads). The tendency of HA constructs to fuse and form integrated bone tissue was demonstrated by the connection between the bone tissue, which was surrounded by a joint fibrous tissue in all the fused constructs (Fig 3B, 3J and 3N). In addition, bone marrow developed along the space between the two adhered tissues (Fig 3N). The implanted constructs were similarly integrated in two out of three cases in collagen constructs; however, the two constructs were attached only via fibrous tissue in the third (Fig 3F).

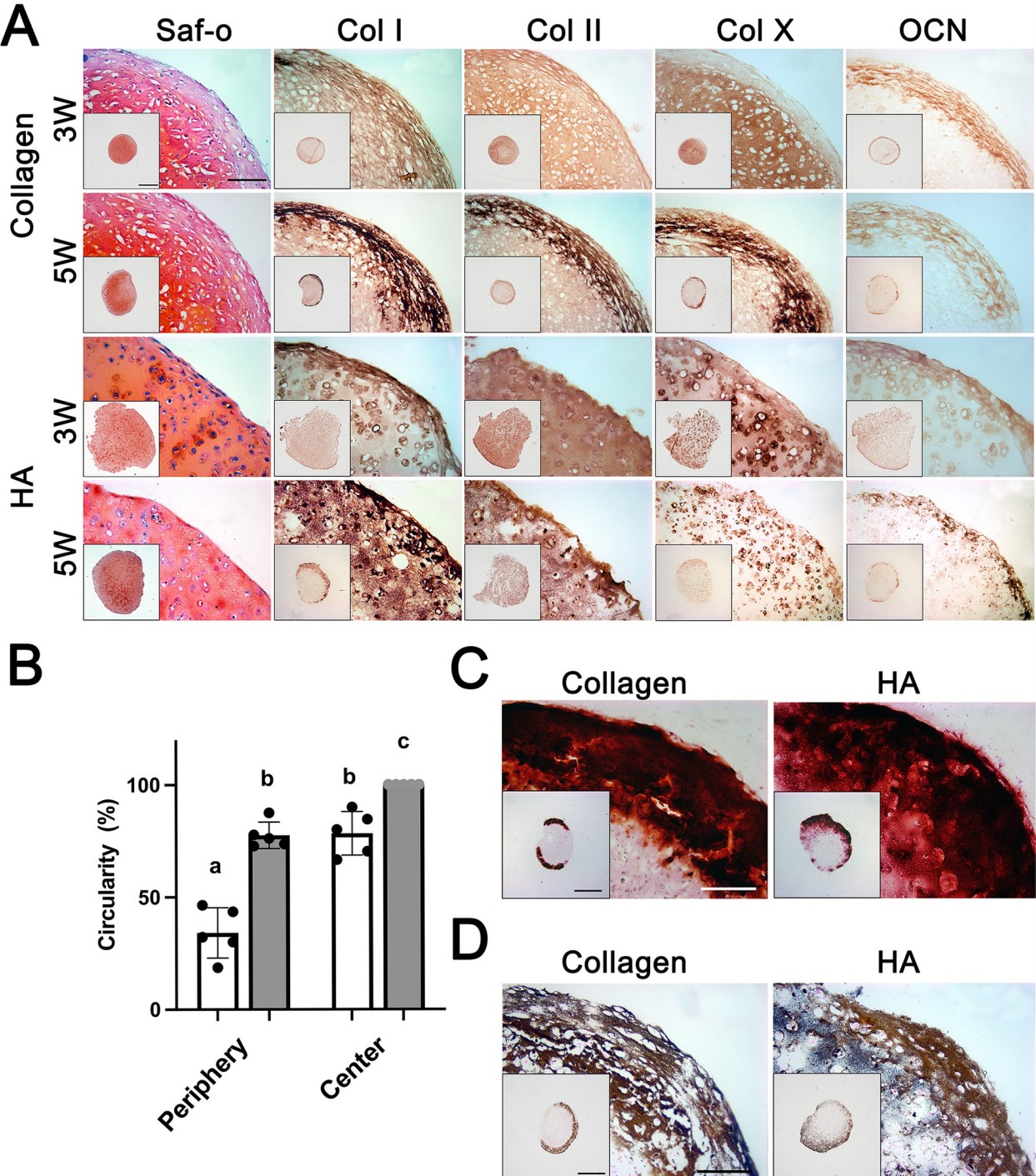

**Fig 2. Histological and immunohistochemical analysis of HA and collagen constructs after in vitro culture.** (A, C, and D) Constructs were stained for sGAG (Safranin O/Fast Green), type I, type II, and type X collagen, osteocalcin (OCN) (A), and calcium (Alizarin Red and Von Kossa, C and D, respectively). All pictures were captured at the same magnification (Scale bar: 200 μm). A low magnification overview of the entire tissues is shown in the insets (Scale bar: 1 mm). (B) The average circularity values of cells at 3 weeks of in vitro culture (n = 5). Groups without a common letter are statistically different (A vs B, P<0.01; B vs C, P<0.05).

**Table 1. Unification rate when multiple constructs were implanted in a single pocket.**

| Hydrogel | Period | Experiments | Adhered | United | % United |
|---|---|---|---|---|---|
| **Collagen** | 4 weeks | 1 | 1 | 1 | |
| | 8 weeks | 4 | 2 | 1 | |
| | Total | 5 | 3 | 2 | 40 |
| **HA** | 4 weeks | 2 | 2 | 2 | |
| | 8 weeks | 3 | 3 | 3 | |
| | Total | 5 | 5 | 5 | 100 |

Two or three constructs were subcutaneously implanted in a single pocket. Constructs were harvested 4 or 8 weeks post-implantation and histologically examined.

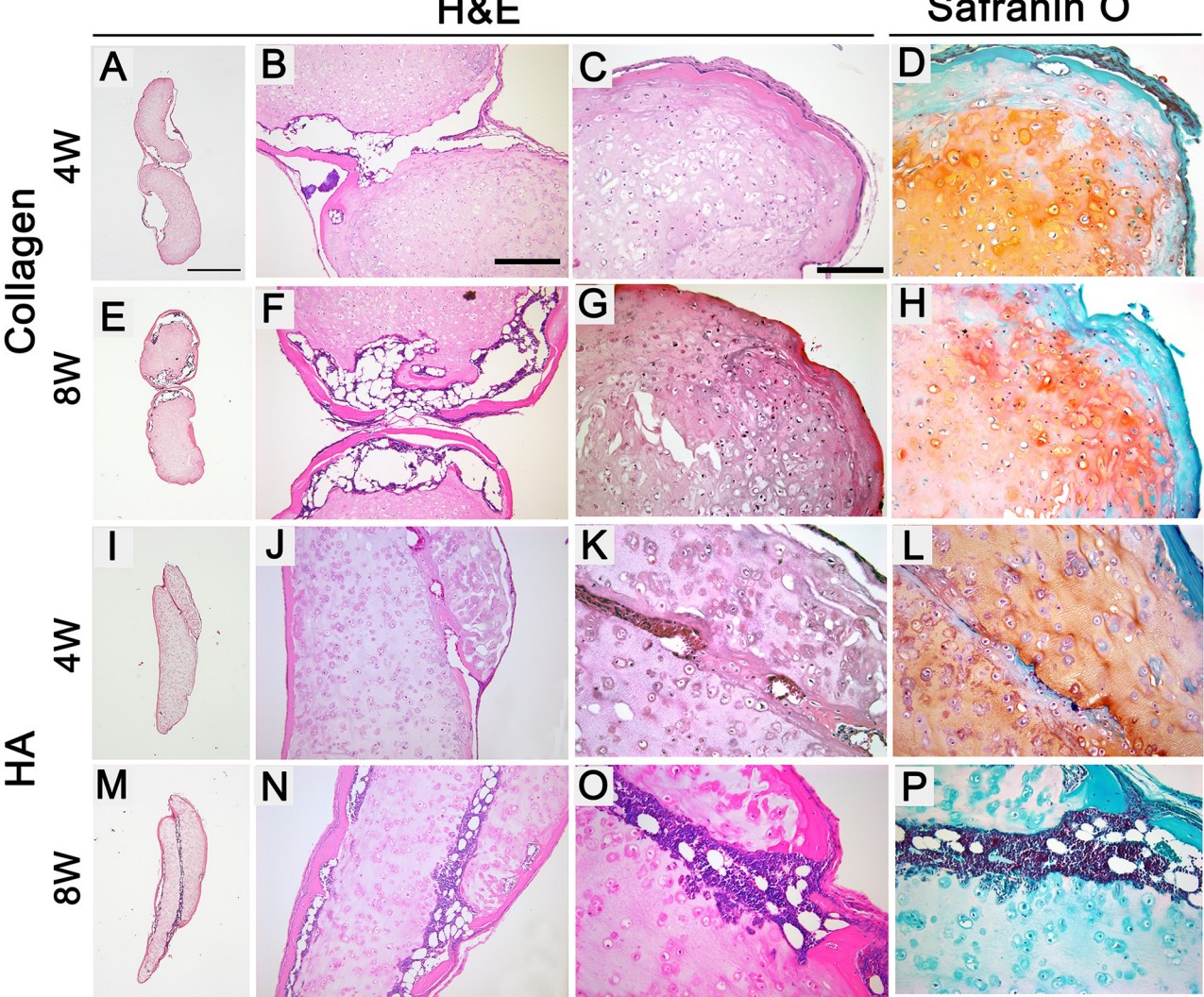

**Fig 3. Histological analysis of HA and collagen constructs post-implantation.** Constructs were stained with H&E (A-C, E-G, I-K, and M-O) and safranin O/Fast Green (D, H, L, and P). Overview of the entire tissues (A, E, I, and M, scale bar: 500 μm). Junction between two constructs (B, F, J, and N, scale bar: 200 μm). (C, D, G, H, K, L, O, and P) Pictures were captured at the same magnification (scale bar, 200 μm). Arrowheads indicate bone marrow developed between the inner cartilage and outer osteoid tissues. Arrows indicate bone marrow components developed along the space between the two adhered tissues and between the inner cartilage and outer osteoid tissues.

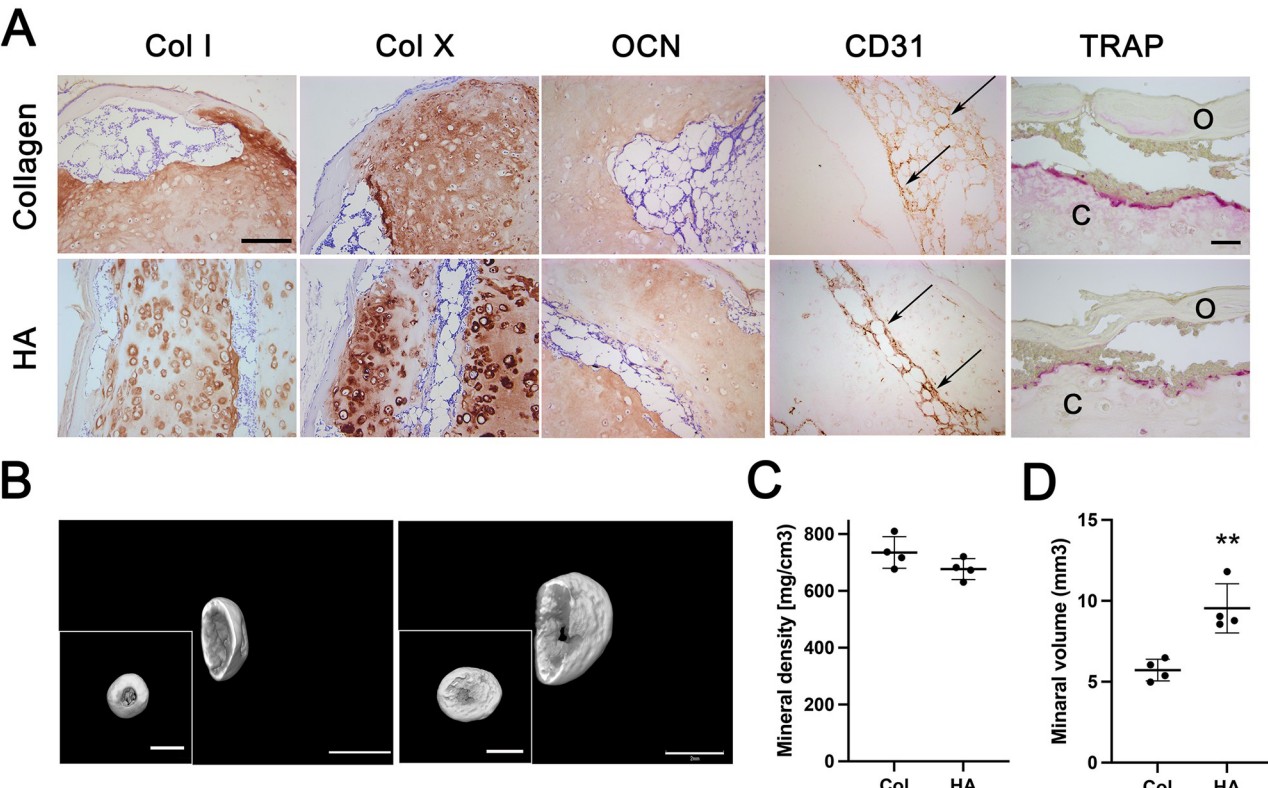

**Fig 4. Immunohistochemical analysis and calcification of HA and collagen constructs post-implantation at 8 weeks.** (A) Constructs were stained for type I, type X, OCN, endothelial cells, and TRAP. Except for TRAP, pictures were captured at the same magnification (scale bar, 200 μm; TRAP scale bar, 50 μm). (B) μCT imaging of collagen (left panel) and HA (right panel) constructs (main and inset image scale bars, 2 mm). (C and D) The total mineral density (C) and volume (D) of HA and collagen constructs (n = 4). ** indicates significant differences; p <0.01. Four constructs were analyzed per group using μCT. Arrows indicate CD31-positive endothelial cells. c, inner cartilage tissue; o, outer osteoid tissue.

**Immunohistochemical analysis.** A loss of chondrogenic phenotype in HA and collagen constructs was indicated by a dramatic increase in type I and X collagen staining intensity, especially in the pericellular space of HA constructs, overlapping with the accumulation of OCN. The newly formed vessels, identified by CD31-positive endothelial cells, penetrated the bone marrow and reached the remodeled inner cartilage area surrounded by TRAP-positive multinucleated cells of the osteoclast lineage, in both HA and collagen constructs (Fig 4A).

**μCT analysis.** At 8 weeks in vivo, deposition of a mineralized matrix in the outer osteoid regions was observed, in both HA and collagen constructs (Fig 4B). Mineral volume was significantly higher in HA constructs than that in the collagen constructs; however, the total mineral density of the new bone was similar between HA and collagen constructs (Fig 4C and 4D).

## Discussion

Using appropriate scaffold materials to facilitate the transition from hypertrophic cartilage to bone is a promising way to scale up MSC-based engineered hypertrophic cartilage grafts, for clinically treating critical-sized bone defects. This study aimed to compare the ability of HA and collagen hydrogels in supporting in vitro differentiation of MSC-based hypertrophic cartilage tissues for promoting endochondral bone formation in vivo. In vitro, both HA and collagen constructs accumulated sGAG and supported collagen synthesis; however, the HA constructs had homogeneous cell morphology and more uniform distribution of sGAG and

collagen proteins. In vivo, both HA and collagen constructs promoted vascularization, endochondral bone formation, and bone marrow development. In addition, HA constructs fused to form an integrated bone.

Regulation of the chondrogenic and hypertrophic phenotypes of MSCs through their interaction with scaffold materials significantly impacts the generation of clinically valid tissue-engineered cartilage grafts. Chondrogenically differentiated cells embedded in HA hydrogels exhibited rounded morphology, which promotes chondrogenesis [18, 27] and supports more uniform chondrogenic differentiation. Unlike HA, collagen interacts with cells through integrin binding sequence (RGD), which could influence the more stretched and diverse cell morphology in collagen constructs. The persistence of this interaction prevents MSCs from further differentiating in the chondrogenic pathway [37]. Therefore, it could support a subpopulation of MSCs that differentiate directly into osteoblasts in response to β-GP in the hypertrophic medium, i.e., IMO [18, 38]. This could explain why hypertrophic marker proteins are preferentially deposited at the periphery under hypertrophic conditions in the collagen constructs.

The fusing of multiple grafts to form a single bone helps extend this ECO-mediated approach to more clinically relevant bone defects. Multiple MSC-based and scaffold-free hypertrophic cartilages applied to a single bone defect indicated that the cartilage grafts integrated with the bone defect. However, the adjacent grafts did not integrate with each other [19], suggesting a requirement for an additional approach to overcome this limitation. The approach in this study using HA as a scaffold may offer one possible solution; compared to that in collagen constructs, multiple grafts of HA constructs had a higher tendency of fusing to form a single bone. The integration tendency and marrow development between the HA constructs could be related to the fact that HA supports uniform chondrogenic differentiation; they promote ECO but inhibit the formation of osteoblast-like cell populations in the peripheral regions, as seen in collagen constructs. ECO is necessary for hematopoietic stem cell niche formation in myelopoiesis [39]; however, IMO lacks processes that promote vascularization and myelopoiesis.

Limitations of this study include the need to accelerate the remodeling rate of the implanted tissues. The implanted engineered tissue remained mainly in a hypertrophic calcified cartilage state even 8 weeks post-implantation. Providing additional channels in hypertrophic cartilage grafts facilitates vascular invasion and mineralization of the graft in vivo [17]. In osteogenesis via the ECO pathways, host-derived cells play a critical role in promoting ECO [13, 40]; therefore, channels created in tissue-engineered hypertrophic cartilage constructs act as conduits for infiltration of host cells and play a role in forming new tissue within the constructs. Alternatively, the implantation of fibrin-encapsulated multiple tiny pellets (μ-pellets) consisting of chondrogenically differentiated MSCs can form integrated bone [41]. Creating such microscale HA constructs would increase the surface area of cartilage tissue surrounded by TRAP-positive osteoclasts, thereby increasing the rate of cartilage tissue remodeling. Promoting engineered graft degradation enhances bone formation [18, 42]. Furthermore, the implantation of many μ-constructs could promote vascularization and bone marrow development by providing increased space for host cells to infiltrate; this would promote integral bone formation with abundant blood vessels. This μ-constructed strategy would accelerate the endochondral ossification process using HA hydrogels, which are excellent for preparing hypertrophic cartilage grafts for bone regeneration.

In conclusion, HA hydrogels are effective for "scaling up" tissue-engineered grafts using fewer cells and gel compared to collagen hydrogels. Furthermore, HA hydrogels have excellent fusion susceptibility and produce integrative bone with vascularization and marrow development between fused grafts. Therefore, improving HA constructs to promote host cell

infiltration and graft remodeling may lead to the development of implantable materials that can further promote vascularization and bone marrow development. With further optimization, this approach could be a promising alternative to current clinical treatments for bone augmentation in maxillofacial and orthopedic surgery.

## Supporting information

**S1 Table. Primer sequences used in quantitative RT-PCR.**
(DOCX)

**S1 Fig. Western blot analysis of HA and collagen constructs at 3 and 5 weeks of in vitro culture.** (A-C) Undifferentiated MSCs expressed Col I but no or very low levels of ACAN and Col II. Except for ACAN in the HA constructs, there was an increased accumulation of Col I, Col II, and ACAN proteins at 5W than 3W. (A) I and II: Uncleaved and cleaved fragments, respectively. (B and C) Unprocessed (pro) or processing intermediate of pro-collagen containing carboxy-pro-peptide (pC). (D and E) GAPDH and α-Tubulin were used as loading controls. The numbers were shown below the bands relative to the Col constructs at 3-week culture. (F) Representative of GAPDH PCR of genomic DNA in the samples loaded into a gel.
(PDF)

**S2 Fig. Sox-9 and MMP-13 expression of HA and collagen constructs at 3 and 5 weeks of in vitro culture.** Constructs were stained for Sox-9 and MMP-13. All pictures were captured at the same magnification (Scale bar: 200 μm). A low magnification overview of the entire tissues is shown in the insets (Scale bar: 1 mm).
(PDF)

**S1 Raw images. Western blot analysis of HA and collagen constructs at 3 and 5 weeks of in vitro culture.**
(PDF)

## Acknowledgments

The authors thank Drs. Ken-Ichi Nakahama and Toshiko Furutera for technical advice on biochemical and histological analysis, respectively. μCT analysis was performed at the Research Core of Tokyo Medical and Dental University (TMDU).

## Author Contributions

**Conceptualization:** Shintaro Yamazaki, Masa-Aki Ikeda.

**Data curation:** Shintaro Yamazaki, Ryoko Hirayama.

**Formal analysis:** Shintaro Yamazaki, Masa-Aki Ikeda.

**Funding acquisition:** Masa-Aki Ikeda.

**Investigation:** Shintaro Yamazaki, Ryoko Hirayama, Masa-Aki Ikeda.

**Methodology:** Yayoi Ikeda, Masa-Aki Ikeda.

**Project administration:** Masa-Aki Ikeda.

**Resources:** Sachiko Iseki, Masa-Aki Ikeda.

**Supervision:** Yayoi Ikeda, Sachiko Iseki, Tetsuya Yoda, Masa-Aki Ikeda.

**Validation:** Yayoi Ikeda, Sachiko Iseki, Tetsuya Yoda, Masa-Aki Ikeda.

**Writing – original draft:** Shintaro Yamazaki, Masa-Aki Ikeda.

**Writing – review & editing:** Masa-Aki Ikeda.

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
