## [Decision Letter · Decision Letter 0]

12 Sep 2022

PONE-D-22-24282Hyaluronic acid hydrogels support to generate integrated bone formation through endochondral ossification in vivo using mesenchymal stem cellsPLOS ONE

Dear Dr. Ikeda,

Thank you for submitting your manuscript to PLOS ONE. After careful consideration, we feel that it has merit but does not fully meet PLOS ONE’s publication criteria as it currently stands. Therefore, we invite you to submit a revised version of the manuscript that addresses the points raised during the review process.

The paper is of interest pending some alterations and more data experiments as suggested by myself and by the reviewers.The Authors in their rebuttal letter must answer point o point to the all the criticisms raised by the reviewers.

We look forward to receiving your revised manuscript.

Kind regards,

Gianpaolo Papaccio, M.D., Ph.D.

Academic Editor

PLOS ONE

Journal Requirements:

2. To comply with PLOS ONE submissions requirements, in your Methods section, please provide additional information on the animal research and ensure you have included details on (1) methods of sacrifice, (2) methods of anesthesia and/or analgesia, and (3) efforts to alleviate suffering.

"The present study was supported by JSPS KAKENHI (grant nos. JP19K10259 and 22K10032). All authors gave their final approval and agree to be accountable for all aspects of the work."

4. Please expand the acronym “JSPS KAKENHI” (as indicated in your financial disclosure) so that it states the name of your funders in full.

"The authors declare that there are no potential conflicts of interest with respect to the research, authorship, and/or publication of this article."

Additional Editor Comments:

Albeit the paper is of interest some alterations are needed:

q-RT PCR data must be confirmed by the relative protein expression in WB.

In addition, Alizarin Red staining in the in vitro samples must be added.

Some recent papers must be added to the References.

Reviewers' comments:

Reviewer's Responses to Questions

**Comments to the Author**

1. Is the manuscript technically sound, and do the data support the conclusions?

Reviewer #1: Yes

2. Has the statistical analysis been performed appropriately and rigorously? 

Reviewer #1: Yes

3. Have the authors made all data underlying the findings in their manuscript fully available?

Reviewer #1: Yes

4. Is the manuscript presented in an intelligible fashion and written in standard English?

Reviewer #1: Yes

5. Review Comments to the Author

Reviewer #1: In this study Authors aimed to compare the ability of HA and collagen hydrogels in supporting in vitro differentiation of MSC-based hypertrophic cartilage tissues for promoting endochondral bone formation in vivo.

The paper is interesting and the experiments are well conducted.

In any case, some changes are needed.

Authors should update the bibliography: there are more recent papers published in the matter (Cells. 2021 Oct 26;10(11):2899. doi: 10.3390/cells10112899; Acta Biomater. 2021 Mar 15;123:364-378. doi: 10.1016/j.actbio.2020.12.056.)

q-RT PCR data should be confirmed by the relative protein expression through WB.

Moreover, it would be interesting to perform also Alizarin Red staining on in vitro samples.

6. PLOS authors have the option to publish the peer review history of their article (what does this mean?). If published, this will include your full peer review and any attached files.

Reviewer #1: No

---

## [Author Response · Author response to Decision Letter 0]

26 Dec 2022

Response to Reviewer ＃1

1. Reviewer’s comment: q-RT PCR data must be confirmed by the relative protein expression in WB. 

Thank you very much for your critical comment. As suggested by the reviewer, we analyzed the relative protein expression of col1, col2, and ACAN by Western blot analysis (S1 Fig). 

First of all、it has been pointed out that the metabolic pathways and cellular structures change considerably when MSCs differentiate into chondrocytes, making it inappropriate to use typically used proteins as loading controls (such as GAPDH). Therefore, in this study, we used DNA content for normalizing sample quantities in Western blotting and the YWHAZ gene as a reference gene for q-RT PCR [35]. 

Consistent with the q-RT PCR data, undifferentiated MSCs expressed Col I but no or very low levels of ACAN and Col II. Although q-RT PCR analysis revealed that Col I, Col II, and ACAN transcript levels decreased from 3W to 5W, except for ACAN in the HA constructs, there was an increased accumulation of Col I, Col II, and ACAN proteins at 5W than 3W, indicating that these proteins continued to accumulate in the constructs, even though their mRNA expression declines during hypertrophic differentiation. 

Sox-9 and MMP13 expression was confirmed by immunohistological staining (S2 Fig), because they were difficult to detect by Western blot analysis. Sox-9expression was observed in cells with fibroblast-like morphology in the cartilage at the margins of the col constructs at 3W. However, no expression was found in the well-differentiated center of the col or HA constructs, which is consistent with the fact that Sox-9 expression peaks at 3 days after MSC induction of chondrogenic differentiation, and after 3 weeks, the expression levels drop to undetectable by Western blotting (Weissenberger et al. BMC Musculoskeletal Disorders (2020) 21:109). Consistent with the q-RT PCR data, MMP-13 protein was not detected at 3W for both constructs; MMP-13 expression at 5W was confined to the marginal area in the collagen constructs, whereas it was observed in the vicinity of chondrocytes throughout the HA construct. Finally, consistent with the q-RT PCR data, the expression of Col X and OCN proteins was detected at both 3W and 5W. 

Collectively, these results demonstrated that the differentiation markers of cartilage and hypertrophic cartilage are expressed in both HA and collagen constructs, although differences in mRNA and protein expression levels are observed. Raw images of Western blot are submitted as ‘S1_raw_images’.

2. Reviewer’s comment: Authors should update the bibliography.

Thank you very much for your helpful comment. We have mentioned and cited the recent papers suggested by the reviewer in Introduction (Lines 62-64) [31,32].

3. Reviewer’s comment: Alizarin Red staining in the in vitro samples must be added. 

As suggested by the reviewer, we have included the data showing Alizarin Red staining in the in vitro samples (Fig 2C). As seen with Von Kossa staining, strong staining was observed limited to the outer edges in both constructs.

---

## [Decision Letter · Decision Letter 1]

23 Jan 2023

Hyaluronic acid hydrogels support to generate integrated bone formation through endochondral ossification in vivo using mesenchymal stem cells

PONE-D-22-24282R1

Dear Dr. Ikeda,

We’re pleased to inform you that your manuscript has been judged scientifically suitable for publication and will be formally accepted for publication once it meets all outstanding technical requirements.

Kind regards,

Gianpaolo Papaccio, M.D., Ph.D.

Academic Editor

PLOS ONE

Additional Editor Comments (optional):

Reviewers' comments:

Reviewer's Responses to Questions

**Comments to the Author**

1. If the authors have adequately addressed your comments raised in a previous round of review and you feel that this manuscript is now acceptable for publication, you may indicate that here to bypass the “Comments to the Author” section, enter your conflict of interest statement in the “Confidential to Editor” section, and submit your "Accept" recommendation.

Reviewer #1: All comments have been addressed

2. Is the manuscript technically sound, and do the data support the conclusions?

Reviewer #1: (No Response)

3. Has the statistical analysis been performed appropriately and rigorously? 

Reviewer #1: (No Response)

4. Have the authors made all data underlying the findings in their manuscript fully available?

Reviewer #1: (No Response)

5. Is the manuscript presented in an intelligible fashion and written in standard English?

Reviewer #1: (No Response)

6. Review Comments to the Author

Reviewer #1: (No Response)

7. PLOS authors have the option to publish the peer review history of their article (what does this mean?). If published, this will include your full peer review and any attached files.

Reviewer #1: No

---

## [Editor Report · Acceptance letter]

25 Jan 2023

PONE-D-22-24282R1 

Hyaluronic acid hydrogels support to generate integrated bone formation through endochondral ossification in vivo using mesenchymal stem cells 

Dear Dr. Ikeda:

I'm pleased to inform you that your manuscript has been deemed suitable for publication in PLOS ONE. Congratulations! Your manuscript is now with our production department. 

Kind regards, 

on behalf of

Prof. Gianpaolo Papaccio 

Academic Editor

PLOS ONE